# Leveraging Contrastive Learning for Enhanced Node Representations in Tokenized Graph Transformers

**Jinsong Chen**[1,2,3], **Hanpeng Liu**[1,3], **John E. Hopcroft**[3,4], **Kun He**[1,3]*

[1]School of Computer Science and Technology, Huazhong University of Science and Technology
[2]Institute of Artificial Intelligence, Huazhong University of Science and Technology
[3]Hopcroft Center on Computing Science, Huazhong University of Science and Technology
[4]Department of Computer Science, Cornell University
{chenjinsong,hanpengliu}@hust.edu.cn,
jeh@cs.cornell.edu, brooklet60@hust.edu.cn

## Abstract

While tokenized graph Transformers have demonstrated strong performance in node classification tasks, their reliance on a limited subset of nodes with high similarity scores for constructing token sequences overlooks valuable information from other nodes, hindering their ability to fully harness graph information for learning optimal node representations. To address this limitation, we propose a novel graph Transformer called GCFormer. Unlike previous approaches, GCFormer develops a hybrid token generator to create two types of token sequences, positive and negative, to capture diverse graph information. And a tailored Transformer-based backbone is adopted to learn meaningful node representations from these generated token sequences. Additionally, GCFormer introduces contrastive learning to extract valuable information from both positive and negative token sequences, enhancing the quality of learned node representations. Extensive experimental results across various datasets, including homophily and heterophily graphs, demonstrate the superiority of GCFormer in node classification, when compared to representative graph neural networks (GNNs) and graph Transformers.

## 1 Introduction

Node classification, a crucial machine learning task in graph data mining, has garnered significant attention recently due to its wide applicability in diverse areas such as social network analysis [24, 35]. Among numerous techniques developed for this task, graph neural networks (GNNs) stand out as the leading architecture due to their exceptional ability to model graph structural data.

Built on the message-passing mechanism [14], GNNs [19, 8, 9, 33, 34] efficiently integrate node and graph topology features to learn informative node representations, effectively preserving both attribute and structural information. However, as research on GNNs progresses, inherent limitations of the message-passing framework, such as over-smoothing [5] and over-squashing [1], have emerged. These limitations hinder GNNs' ability to capture long-range dependencies in graphs, ultimately constraining their potential for node classification.

Recently, the emerging graph Transformer has attracted great attention in the field of graph representation learning. The crux of this approach is to leverage the Transformer architecture to learn node representations from the input graph. Benefiting from the self-attention mechanism in Transformer, graph Transformers [45, 17, 6, 7, 50] can effectively capture the long-range dependencies in graphs. Serving as a new deep learning-based technique for graphs, graph Transformers have showcased

---

*Corresponding author.

38th Conference on Neural Information Processing Systems (NeurIPS 2024).

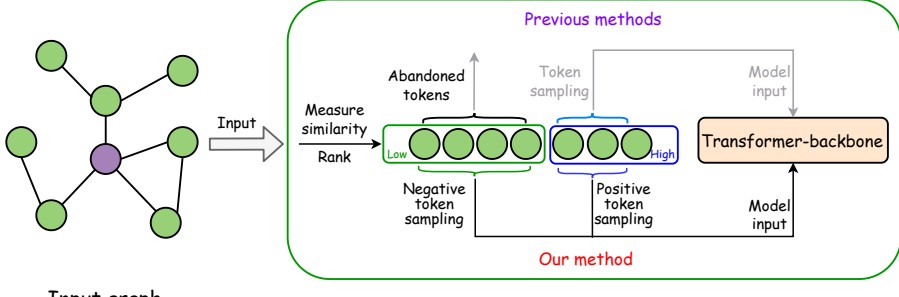

Figure 1: A toy example to illustrate the difference of the token generator between the token generator in our method and that used in the previous node tokenized graph Transformers. Previous methods only sample nodes with high similarity to construct token sequences. In contrast, our method introduces both positive and negative token sampling to preserve information carried by diverse nodes in the graph.

remarkable performance in the node classification task. In this study, We roughly divide the existing graph Transformers designed for node classification into two categories according to the model architecture: GNN-based graph Transformers and tokenized graph Transformers.

GNN-based graph Transformers [30, 42, 41, 23, 27] utilize a hybrid framework that merges Transformer layers with GNN-style modules to learn node representations. However, this approach may constrain the modeling capacity of the Transformer architecture due to the deeply coupled design of the Transformer and GNN layers. A recent study [44] also theoretically proves that directly applying Transformer to calculate the attention scores of all node pairs could cause the over-globalizing problem, which causes the model to overly rely on global information, negatively affecting the model's performance.

In contrast, tokenized graph Transformers [52, 50, 7, 13] initially generate token sequences for each node and only calculate attention scores between tokens within the token sequence, naturally avoiding the over-globalizing issue. These token sequences are then processed by a Transformer-based backbone to learn node representations. This mechanism allows the Transformer to flexibly extract informative node representations based on the input token sequences, demonstrating impressive performance in node classification. Note that, tokenized graph Transformers focus on building token sequences for each target node as model inputs, which is different from TokenGT [18] that transforms all elements in graphs as tokens.

Token generation is a crucial step in tokenized graph Transformers, where node [50] and neighborhood [7] elements form the core of the token sequences. While neighborhood tokens primarily preserve local topology features [13], node tokens can capture a broader range of graph information, including long-range dependencies and intrinsic graph properties (*e.g.*, homophily and heterophily). These advantages allow graph Transformers built on node-oriented token sequences [52, 50, 13] to learn more informative node representations, compared to those based on neighborhood-oriented token sequences.

In this study, we observe that the techniques employed by existing tokenized graph Transformers for generating node-orient token sequences could be summarized as a two-step method. First, they estimate the node similarity matrix according to node information across various feature spaces, such as topology features [52] and attribute features [50, 13]. They then sample a fixed number of nodes with high similarity scores from the generated similarity matrix to construct the input token sequence for a target node. As depicted in Figure 1, only a small subset of nodes is considered, while other nodes are excluded during the training stage.

Compared to sampled nodes which capture the commonality with the target node, these abandoned nodes preserve the disparity, which is also valuable for learning distinguishable node representations. A previous study [3] has proved that leveraging the information from dissimilar nodes aids the

learning of node representations. Nevertheless, existing tokenized graph Transformers can not comprehensively utilize both similar and dissimilar nodes to learn the representation of the target node, inevitably limiting the model performance for node classification. Hence, a natural question arises: *How should we design a new graph Transformer to comprehensively and effectively leverage diverse nodes in graphs to learn distinguishable node representations?*

To answer this question, we propose a new method called Graph Contrastive Transformer (GCFormer). Unlike previous graph Transformers, GCFormer first introduces a novel token sequence generator that produces both positive and negative token sequences for each node in different feature spaces. In this way, various graph information carried by node tokens can be carefully preserved in different types of token sequences. Then, GCFormer develops a new Transformer-based backbone tailored for effectively learning node representations from the generated positive and negative token sequences. Finally, GCFormer leverages the contrastive learning to comprehensively utilize the tokens in both positive and negative sequences to further enhance the quality of learned node representations.

The main contributions of this paper are summarized as follows:

- We develop a new token sequence generator that can generate different types of token sequences in terms of positive and negative node tokens for each target node to preserve various graph information.

- We propose a new graph Transformer GCFormer that formulates a Transformer-based backbone and leverages the contrastive learning to comprehensively learn node representations from positive and negative token sequences.

- We conduct extensive experiments on both homophily and heterophily graphs to validate the effectiveness of the proposed method. Experimental results demonstrate the superiority of GCFormer in node classification compared to representative GNNs and graph Transformers.

## 2 Related Work

In this section, we first introduce recent studies of graph Transformers for node classification. We then briefly review studies about contrastive learning on graphs.

### 2.1 Graph Transformer

We categorize existing graph Transformers for node classification into GNN-based methods and tokenized methods. The former [30, 41, 42, 23, 27] combines the Transformer layers with GNN-style modules to learn node representations. GraphGPS [41], one of the representative approaches, incorporates various linear Transformers, such as Reformer [20] and BigBird [47], and GNN layers [19] into a unified framework for graph representation learning. However, these approaches require performing the attention calculation on all node pairs, which can lead to what is known as the over-globalizing problem. A recent study [44] provides both empirical evidence and theoretical analysis to show that calculating attention scores for all nodes can cause the model to overly rely on global information, which can negatively affect the model's performance in node classification tasks.

In contrast, tokenized methods purely depend on the Transformer architecture. The key idea is to generate token sequences for each node from the input graph, which are then fed to Transformer to learn node representations. Node-based [52, 50, 13] and neighborhood-based [7, 11, 13] token generators have been developed to generate various token sequences for nodes. Node-based token generators first calculate the similarity of nodes according to node features such as attribute features [50], then sample nodes with high similarity scores as tokens of the input sequence. While neighborhood-based token generators [7] aggregate the features of multi-hop neighborhoods and further transform them into tokens to construct the token sequence. Compared to neighborhood-based tokens, node-based tokens can express more complex graph information, such as long-range dependencies, which are more suitable for learning informative node representations.

Different from previous node token-based graph Transformers that only consider nodes with high similarity, our proposed GCFormer generates both positive and negative token sequences from all nodes in the graph. Various graph information carried by diverse nodes in two types of token sequences enables GCFormer to learn more distinguishable node representations, leading to superior performance.

## 2.2 Contrastive Learning on Graphs

Graph contrastive learning (GCL) [36, 54, 46, 31, 49] aims to introduce the contrastive learning mechanism into GNNs to learn informative representations of graphs. Most of GCL approaches share a similar framework that first performs graph augmentation techniques to generate various features of different graph views and then applies the contrastive loss on these generated views to learn graph representations [31]. Recent studies [32, 53, 48, 51] attempt to introduce contrastive learning into graph Transformers. However, these methods require the entire graph as the model input [51, 48] or need to combine GNN-based modules with tailored graph augmentation strategies [32, 53], which are hard to directly apply on tokenized graph Transformers in the node classification task.

Our proposed GCFormer develops a new token generator to generate both positive and negative token sequences for each node without any data augmentations. With the dedicated Transformer-based backbone, GCFormer can effectively leverage the contrastive learning to comprehensively learn informative node representations from two types of token sequences.

## 3 Preliminaries

### 3.1 Node Classification

Consider an attributed graph $\mathcal{G} = (V, E)$, where $V$ and $E$ are the node and edge sets, respectively. We have the corresponding adjacency matrix $\mathbf{A} \in \mathbb{R}^{n \times n}$, where $n$ is the number of nodes. For arbitrary two nodes $v_i$ and $v_j$, $\mathbf{A}_{ij} = 1$ only if $e_{ij} \in E$. The diagonal degree matrix $\mathbf{D} \in \mathbb{R}^{n \times n}$ is represented as $\mathbf{D}_{ii} = \sum_{j=1}^{n} \mathbf{A}_{ij}$. The normalized version of the adjacency matrix with self-loops is represented as $\hat{\mathbf{A}} = (\mathbf{D} + \mathbf{I})^{-1/2}(\mathbf{A} + \mathbf{I})(\mathbf{D} + \mathbf{I})^{-1/2}$, where $\mathbf{I}$ denotes the identity matrix. Nodes in $\mathcal{G}$ are associated with attribute feature vectors, assembled into an attribute feature matrix denoted as $\mathbf{X}^a \in \mathbb{R}^{n \times d}$ where $d$ is the dimension of the feature vector. The node label matrix $\mathbf{Y} \in \mathbb{R}^{n \times c}$, where $c$ is the label count, consists of rows that are one-hot vectors encoding the label of each node. Each row in $\mathbf{Y}$ is a one-hot vector representing the label information of the corresponding node. Given a subset of nodes with known labels $V_l$, the objective of node classification is to infer the labels for the remaining nodes in the set $V - V_l$.

### 3.2 Transformer

Transformer stands as a notable model in deep learning, built upon the Encoder-Decoder architecture. This brief overview focuses on the Transformer layer, a pivotal component of the model. Each Transformer layer is composed of two essential parts: Multi-Head Self-Attention (MSA) and Feed-Forward Networks (FFN).

MSA harnesses multiple attention heads, employing the self-attention mechanism to refine the representations of input entities. Given the input feature matrix $\mathbf{H} \in \mathbb{R}^{n \times d_i n}$, the calculation of the $i$-th attention head is as follows:

$$\text{head}_i(\mathbf{H}) = \text{Softmax}(\frac{\mathbf{Q}\mathbf{K}^{\text{T}}}{\sqrt{d_k}})\mathbf{V}, \tag{1}$$

where $\mathbf{Q} = \mathbf{H}\mathbf{W_Q}$, $\mathbf{K} = \mathbf{H}\mathbf{W_K}$ and $\mathbf{V} = \mathbf{H}\mathbf{W_V}$. $\mathbf{W_Q} \in \mathbb{R}^{d_{in} \times d_k}$, $\mathbf{W_K} \in \mathbb{R}^{d_{in} \times d_k}$ and $\mathbf{W_V} \in \mathbb{R}^{d_{in} \times d_v}$ are learnable parameter matrices. The output of MSA with $m$ attention heads is calculated as:

$$\mathbf{H}' = (\text{head}_1 || \text{head}_2 || \cdots || \text{head}_m)\mathbf{W_O}, \tag{2}$$

where $||$ denotes the vector concatenation operation and $\mathbf{W_O}$ is the learnable matrix.

FFN, comprised of two linear layers enveloping a nonlinear activation function, is defined as:

$$\mathbf{H}' = \text{Linear}(\sigma(\text{Linear}(\mathbf{H}))), \tag{3}$$

where $\text{Linear}(\cdot)$ indicates a linear layer, and $\sigma(\cdot)$ symbolizes the nonlinear activation function.

## 4 Method

In this section, we detail our proposed GCFormer. First, we introduce the hybrid token generator, which produces both positive and negative token sequences for each node. Then, we introduce the

tailored Transformer-based backbone for extracting node representations from the generated token sequences. Finally, we introduce how to integrate contrastive learning into GCFormer to enhanced node representations.

## 4.1 Hybrid Token Generator

The proposed hybrid token generator contains two steps: similarity estimating and node sampling. The critical operation of similarity estimating is to calculate the similarity score matrix $\mathbf{S} \in \mathbb{R}^{n \times n}$ of all node pairs. Obviously, different node features lead to different score matrices, describing node pairs' relations in different feature spaces. To preserve the complex relations of nodes in the graph, besides the attribute-aware feature matrix $\mathbf{X}^a$, we construct the topology-aware feature matrix $\mathbf{X}^t$:

$$\mathbf{X}^t = \hat{\mathbf{A}}^k \mathbf{X}^a, \tag{4}$$

where $k$ is the propagation step. $\mathbf{X}^t$ preserves the local topology feature within the $k$-hop neighborhood for each node, which is the essential information to characterize the node property on the graph [7, 16].

Then, we utilize the cosine similarity to calculate the similarity score $\mathbf{S}^a \in \mathbb{R}^{n \times n}$ and $\mathbf{S}^t \in \mathbb{R}^{n \times n}$ based on the node feature matrices $\mathbf{X}^a$ and $\mathbf{X}^t$, respectively. Given a node pair $(v_i, v_j)$, the similarity scores in the attribute feature space $\mathbf{S}^a_{ij}$ and topology feature space $\mathbf{S}^t_{ij}$ are calculated as follows:

$$\mathbf{S}^a_{ij} = \frac{\mathbf{X}^a_i \cdot \mathbf{X}^{a^{\mathrm{T}}}_j}{|\mathbf{X}^a_i||\mathbf{X}^a_j|}, \quad \mathbf{S}^t_{ij} = \frac{\mathbf{X}^t_i \cdot \mathbf{X}^{t^{\mathrm{T}}}_j}{|\mathbf{X}^t_i||\mathbf{X}^t_j|}. \tag{5}$$

After estimating the similarity scores of all node pairs, GCFormer then conducts a two-stage sampling process involving positive token sampling and negative node sampling to generate the token sequences. Here, we introduce the sampling process based on the attribute similarity matrix $\mathbf{S}^a$ for a simplified description. For a given target node $v_i$, in the positive token sampling stage, we adopt the top-$k$ strategy to select nodes to construct the positive token sequence:

$$V^{a,p}_i = \{v_j | v_j \in \mathrm{Top}(\mathbf{S}^a_i)\}, \tag{6}$$

where $\mathrm{Top}(\cdot)$ denotes the top-$k$ sampling function and $V^{a,p}_i$ denotes the positive token sequence with length $p_k$. As for the negative token sampling stage, we have the set of rest nodes for $v_i$ after positive token sampling $V^{a,r}_i = V - V^{a,p}_i$. In this paper, we regard all nodes in $V^{a,r}_i$ as the negative samples since their similarity scores are below the threshold of top-$k$ selection. Then, we apply the sampling function to sample nodes from $V^{a,r}_i$ to construct the negative token sequence for $v_i$:

$$V^{a,n}_i = \{v_j | v_j \in \mathrm{Sample}(V^{a,r}_i)\}, \tag{7}$$

where $\mathrm{Sample}(\cdot)$ denotes an arbitrary sampling function. Here, we use uniform sampling for computing efficiency. $V^{a,n}_i$ denotes the negative token sequence with length $n_k$.

Following the same sampling process, we can obtain positive and negative token sequences $V^{t,p}_i$ and $V^{t,n}_i$ based on the topology similarity matrix $\mathbf{S}^t$. The constructed positive and negative token sequences not only capture node relations in different feature spaces but also comprehensively extract valuable information from all nodes on the graph.

## 4.2 Transformer-based Backbone

GCFormer formulates a Transformer-based backbone to effectively learn node representations from positive and negative token sequences. For a node $v_i$, we first combine itself with generated positive and negative token sequences to construct the model input, $\mathbf{H}^{a,i^o} \in \mathbb{R}^{(1+p_k+n_k) \times d} = \{\mathbf{X}_i, \mathbf{X}_p, \mathbf{X}_n | v_p \in V^{a,p}_i, v_n \in V^{a,n}_i\}$ and $\mathbf{H}^{t,i^o} \in \mathbb{R}^{(1+p_k+n_k) \times d} = \{\mathbf{X}^t_i, \mathbf{X}^t_p, \mathbf{X}^t_n | v_p \in V^{t,p}_i, v_n \in V^{t,n}_i\}$. Note that we utilize the generated $\mathbf{X}^t$ to construct the model input of topology-aware token sequences. In this way, the topology features can be carefully preserved in the model input $\mathbf{H}^{t,i^o}$, exhibiting significant differences with previous methods [52, 50, 13] that utilize the attribute features to construct topology-aware token sequences. Following previous studies [8, 7, 13], we leverage projection layers to obtain the initial input:

$$\mathbf{H}^{a,i} = \mathbf{H}^{a,i^o} \mathbf{W}^a, \quad \mathbf{H}^{t,i} = \mathbf{H}^{t,i^o} \mathbf{W}^t, \tag{8}$$

where $\mathbf{W}^a \in \mathbb{R}^{d \times d_0}$ and $\mathbf{W}^t \in \mathbb{R}^{d \times d_0}$ denote the parameter matrices of the projection layers.

Given the model input $\mathbf{H}^{a,i}$ of the node $v_i$, GCFormer first separates the negative tokens from $\mathbf{H}^{a,i}$, resulting in two parts: $\mathbf{P}^{a,i^{(0)}} \in \mathbb{R}^{(1+p_k) \times d_0}$ and $\mathbf{N}^{a,i^{(0)}} \in \mathbb{R}^{n_k \times d_0}$. Next, GCFormer adds a virtual token with learnable features into $\mathbf{N}^{a,i^{(0)}}$ as the first token to facilitate extracting valuable information from negative tokens. Then, GCFormer adopts standard Transformers layers to learn node representations from $\mathbf{P}^{a,i^{(0)}}$ and $\mathbf{N}^{a,i^{(0)}}$:

$$\mathbf{P}^{a,i^{(l)'}} = \text{MSA}(\mathbf{P}^{a,i^{(l-1)}}) + \mathbf{P}^{a,i^{(l-1)}}, \quad \mathbf{P}^{a,i^{(l)}} = \text{FFN}(\mathbf{P}^{a,i^{(l)'}}) + \mathbf{P}^{a,i^{(l)'}}, \tag{9}$$

$$\mathbf{N}^{a,i^{(l)'}} = \text{MSA}(\mathbf{N}^{a,i^{(l-1)}}) + \mathbf{N}^{a,i^{(l-1)}}, \quad \mathbf{N}^{a,i^{(l)}} = \text{FFN}(\mathbf{N}^{a,i^{(l)'}}) + \mathbf{N}^{a,i^{(l)'}}, \tag{10}$$

where $\text{MSA}(\cdot)$ and $\text{FFN}(\cdot)$ denote the multi-head self-attention and feed-forward networks.

Through several Transformer layers, the corresponding $\mathbf{P}^{a,i} \in \mathbb{R}^{(1+p_k) \times d_{out}}$ and $\mathbf{N}^{a,i} \in \mathbb{R}^{(1+n_k) \times d_{out}}$ contains information extracted from positive and negative token sequences, respectively. To effectively fuse information from different types of token sequences, inspired by signed attention mechanism in previous approaches [3, 10], we develop the following readout function:

$$\mathbf{H}^{a,i} = \mathbf{P}_0^{a,i} - \mathbf{N}_0^{a,i}, \tag{11}$$

where $\mathbf{H}^{a,i} \in \mathbb{R}^{1 \times d_{out}}$ denote the node representation of $v_i$ extracted from the attribute-aware token sequence.

The rationale of Equation 11 is that the representations $\mathbf{P}_0^{a,i}$ (the target node) and $\mathbf{N}_0^{a,i}$ (the virtual node) contain the learned information from positive and negative token sequences, respectively. The desired representation of $v_i$ should be far away from the representations of negative tokens in the hidden feature space since there is a high probability that they belong to different labels. While the signed aggregation operation can enforce $\mathbf{H}^{a,i}$ to be dissimilar with the representations of negative tokens according to the previous study [3, 10].

We can also obtain the representation $\mathbf{H}^{t,i} \in \mathbb{R}^{1 \times d_{out}}$ extracting from the topology-aware token sequence $\mathbf{H}^{t,i^o}$ via the same operation. Considering the contributions of attribute information and topology information vary on different graphs, we develop a weighted fusion strategy to obtain the final representation $\mathbf{Z}^i$:

$$\mathbf{Z}^i = \alpha \cdot \mathbf{H}^{a,i} + (1 - \alpha) \cdot \mathbf{H}^{t,i}, \tag{12}$$

where $\alpha \in [0, 1]$ is a hyper-parameter to determine the contributions of attribute information and topology information to the final representation.

### 4.3 Integrating Contrastive Learning

Though Equation 11 leverages information of negative tokens to learn node representation, it fails to directly model relations between the target node and its negative tokens. To this end, we introduce the contrastive learning loss [15] to fully utilize negative tokens for enhanced node representations. For a node $v_i$, the contrastive learning loss is calculated as follows:

$$\mathcal{L}_{cl}(v_i) = -\log \frac{\exp(\mathbf{P}_0^{a,i} \cdot \hat{\mathbf{P}}^{a,i^{\text{T}}}/\tau)}{\sum_{j=1}^{n_k} \exp(\mathbf{P}_0^{a,i} \cdot \mathbf{N}_j^{a,i^{\text{T}}}/\tau)} - \log \frac{\exp(\mathbf{P}_0^{t,i} \cdot \hat{\mathbf{P}}^{t,i^{\text{T}}}/\tau)}{\sum_{j=1}^{n_k} \exp(\mathbf{P}_0^{t,i} \cdot \mathbf{N}_j^{t,i^{\text{T}}}/\tau)}, \tag{13}$$

where $\hat{\mathbf{P}}^{a,i} = \frac{1}{p_k} \sum_{j=1}^{p_k} \mathbf{P}_j^{a,i}$ and $\hat{\mathbf{P}}^{t,i} = \frac{1}{p_k} \sum_{j=1}^{p_k} \mathbf{P}_j^{t,i}$. $\tau$ is a temperature hyper-parameter. Equation 13 enforces the representation of the target node to be close to the central representation of all positive tokens and away from all negative samples, which promotes learning distinguishable node representations, beneficial for downstream classification tasks. We further adopt the Cross-entropy loss for node classification:

$$\mathcal{L}_{ce} = -\sum_{i \in V_l} \mathbf{Y}_i \ln \hat{\mathbf{Y}}_i, \hat{\mathbf{Y}}_i = \text{MLP}(\mathbf{Z}^i), \tag{14}$$

where $\text{MLP}(\cdot)$ denotes the Multilayer Perceptron-based classifier. Hence, the overall loss function of GCFormer is as follows:

$$\mathcal{L} = \mathcal{L}_{ce} + \beta \cdot \mathcal{L}_{cl}, \tag{15}$$

where $\beta$ is the coefficient for the contrastive learning term.

Table 1: Comparison of all models in terms of mean accuracy $\pm$ stdev (%). The best results appear in **bold**. The second results appear in underline.

| Dataset | Photo | ACM | Comuter | Corafull | BlogCatalog | UAI2010 | Flickr | Romanempire |
|---|---|---|---|---|---|---|---|---|
| $H(\mathcal{G})$ | 0.83 | 0.82 | 0.78 | 0.57 | 0.40 | 0.36 | 0.24 | 0.05 |
| APPNP | $93.00_{\pm0.55}$ | $93.00_{\pm0.55}$ | $\underline{91.31}_{\pm0.29}$ | $63.37_{\pm0.04}$ | $\underline{94.77}_{\pm0.19}$ | $76.41_{\pm0.47}$ | $84.66_{\pm0.31}$ | $52.96_{\pm0.35}$ |
| SGC | $93.74_{\pm0.07}$ | $93.24_{\pm0.49}$ | $88.90_{\pm0.11}$ | $62.77_{\pm0.19}$ | $72.61_{\pm0.07}$ | $69.87_{\pm0.17}$ | $47.48_{\pm0.40}$ | $34.42_{\pm0.77}$ |
| GPRGNN | $94.57_{\pm0.44}$ | $93.42_{\pm0.20}$ | $90.15_{\pm0.34}$ | $69.08_{\pm0.11}$ | $94.36_{\pm0.29}$ | $\underline{76.94}_{\pm0.64}$ | $85.91_{\pm0.51}$ | $67.06_{\pm0.27}$ |
| FAGCN | $94.06_{\pm0.03}$ | $93.37_{\pm0.24}$ | $83.17_{\pm1.81}$ | $56.61_{\pm2.94}$ | $79.92_{\pm4.39}$ | $72.17_{\pm1.57}$ | $82.03_{\pm0.40}$ | $48.21_{\pm3.15}$ |
| ACM-GCN | $94.56_{\pm0.21}$ | $93.04_{\pm1.28}$ | $85.19_{\pm2.26}$ | $65.11_{\pm1.98}$ | $94.53_{\pm0.53}$ | $76.87_{\pm1.42}$ | $83.85_{\pm0.73}$ | $63.35_{\pm1.80}$ |
| SGFormer | $92.93_{\pm0.12}$ | $93.79_{\pm0.34}$ | $81.86_{\pm3.82}$ | $64.62_{\pm1.20}$ | $94.33_{\pm0.19}$ | $57.98_{\pm3.95}$ | $61.05_{\pm0.68}$ | $41.31_{\pm0.51}$ |
| ANS-GT | $94.88_{\pm0.23}$ | $\underline{93.92}_{\pm0.21}$ | $89.58_{\pm0.28}$ | $67.94_{\pm0.21}$ | $91.93_{\pm0.31}$ | $74.16_{\pm0.71}$ | $85.94_{\pm0.25}$ | $73.95_{\pm0.32}$ |
| Specformer | $95.22_{\pm0.13}$ | $93.63_{\pm1.94}$ | $85.47_{\pm1.44}$ | $69.18_{\pm0.24}$ | $94.21_{\pm0.23}$ | $73.06_{\pm0.77}$ | $86.55_{\pm0.40}$ | $63.69_{\pm0.61}$ |
| VCR-Graphormer | $95.13_{\pm0.24}$ | $93.24_{\pm0.31}$ | $90.14_{\pm0.43}$ | $68.96_{\pm0.28}$ | $93.92_{\pm0.37}$ | $75.78_{\pm0.69}$ | $86.23_{\pm0.74}$ | $74.76_{\pm0.83}$ |
| GraphGPS | $93.79_{\pm0.32}$ | $93.31_{\pm0.26}$ | $89.21_{\pm0.28}$ | $62.08_{\pm0.35}$ | $94.35_{\pm0.52}$ | $75.44_{\pm0.48}$ | $83.61_{\pm0.57}$ | $68.29_{\pm0.92}$ |
| NAGphormer | $\underline{95.47}_{\pm0.29}$ | $93.32_{\pm0.30}$ | $90.79_{\pm0.45}$ | $\underline{69.34}_{\pm0.52}$ | $94.42_{\pm0.63}$ | $76.36_{\pm1.12}$ | $\underline{86.85}_{\pm0.85}$ | $\underline{74.94}_{\pm0.52}$ |
| GCFormer | $\mathbf{95.65}_{\pm0.41}$ | $\mathbf{94.32}_{\pm0.47}$ | $\mathbf{92.09}_{\pm0.21}$ | $\mathbf{69.53}_{\pm0.35}$ | $\mathbf{96.03}_{\pm0.44}$ | $\mathbf{77.57}_{\pm0.86}$ | $\mathbf{87.90}_{\pm0.45}$ | $\mathbf{75.38}_{\pm0.68}$ |

# 5 Experiments

## 5.1 Experimental Setup

We briefly introduce the experimental setup including datasets, baselines and parameter settings. Detailed information is provided in Appendix A due to the space limitation.

**Datasets.** We adopt eight widely used datasets, including four homophily and four heterophily graphs: Photo [7], ACM [37], Computer [7], Corafull [4], BlogCatalog [28], UAI2010 [38], Flickr [28] and Romanempire [29]. The edge homophily ratio [22] $H(\mathcal{G}) \in [0, 1]$ is adopted to evaluate the graph's homophily level. $H(\mathcal{G}) \to 1$ means strong homophily, while $H(\mathcal{G}) \to 0$ means strong heterophily. Statistics of datasets are summarized in Appendix A. Following the settings of previous studies [41, 42], we randomly choose 50% of each label as the training set, 25% as the validation set, and the rest as the test set.

**Baselines.** We adopt eleven powerful approaches on node classification as baselines, including GNNs and graph Transformers: APPNP [21], SGC [40], GPRGNN [12], FAGCN [3], ACM-GCN [26], SGFormer [42], ANS-GT [50], Specformer [2], VCR-Graphormer [13], GraphGPS [30] [1] and NAGphormer [7]. The first five are representative GNNs and others are recent graph Transformers.

**Parameter settings.** For baselines, referring to recommended settings in their official implementations, we perform hyper-parameter tuning for all models. For GCFormer, we try the dimension of hidden representations in $\{128, 256, 512\}$, number of layers in $\{1, 2, 3\}$, learning rate in $\{0.01, 0.005, 0.001\}$, dropout rate in $\{0.1, 0.3, 0.5\}$. The training process is early stopped within 50 epochs and parameters are optimized using AdamW [25].

## 5.2 Performance Comparison

To evaluate the model performance in node classification, we run each model with different random seeds on datasets and report the average value of accuracy and the corresponding standard deviation.

Table 1 reports the results. We can observe that GCFormer achieves the best performance on all datasets, indicating the superiority of GCFormer on the node classification task. Specifically, GCFormer beats recent tokenized graph Transformers on all datasets, especially ANS-GT which is the representative method of node token sequence-based graph Transformers. This is because that GCFormer generates both positive and negative token sequences for each node, which preserve both commonality and disparity between node features. In addition, the tailored Transformer-based backbone and contrastive learning enable GCFormer to comprehensively learn distinguishable node representations from different types of token sequences, further enhancing the performance in the node classification task. Moreover, we also find graph Transformer-based baselines achieve higher accuracy values than GNN-based baselines on over half of datasets. This is because graph Transformers can

---

[1] Due to the various implementations of GraphGPS, here we only report the best combination. Detailed results of all combinations can refer to Appendix C.

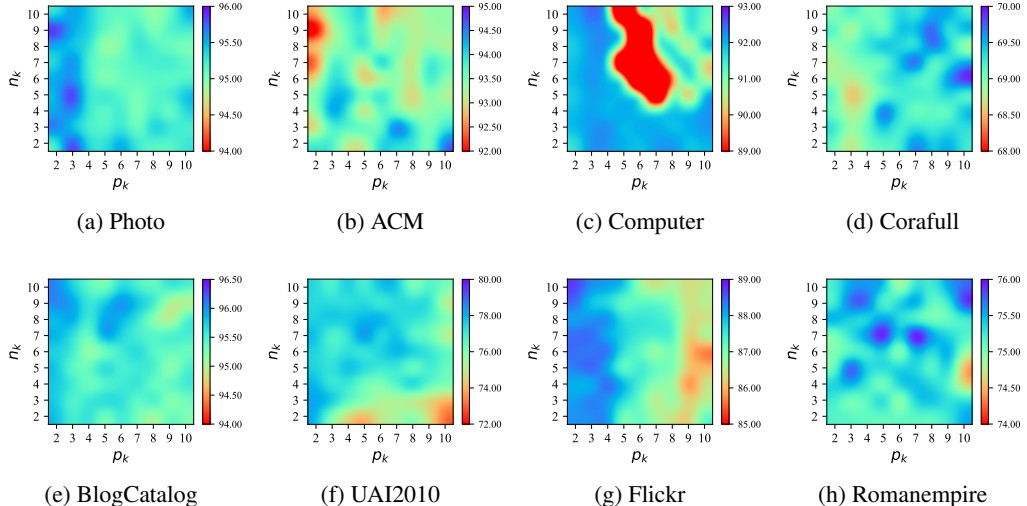

Figure 2: Performance of GCFormer with different sampling sizes on all datasets.

effectively preserve various graph information, such as local topology features [7] and long-range dependencies [50], revealing the potential of graph Transformers in graph mining tasks.

## 5.3 Parameter Sensitivity Analysis

The token sampling size and the aggregation weight $\alpha$ in Equation 12 are key parameters in GCFormer. The former determines the model input and the latter controls the learning of final node representations from different feature spaces. Here, we conduct experiments to analyze the influence of these parameters on model performance.

**Analysis of token sampling sizes.** To analyze the influence of different sampling sizes on model performance, we vary $p_k$ and $n_k$ in $\{2, 3, \ldots, 10\}$ where $p_k$ and $n_k$ are the lengths of positive token sequences and negative token sequences. Figure 2 shows the changes in model performance across all datasets. Generally speaking, a large sampling size of negative tokens can lead to competitive model performance. For instance, $n_k$ over six can enable GCFormer to achieve high accuracy on almost all datasets except ACM. This is because a large value of $n_k$ is more conducive to preserving the disparity between target nodes and negative node tokens, leading to more distinguishable node representations. This phenomenon also indicates that introducing negative tokens can effectively enhance the performance of tokenized graph Transformers in node classification. In addition, GCFormer is relatively sensitive to $n_p$. Half of the datasets, such as Photo and BlogCatalog, require a small value of $n_p$ to achieve competitive performance. While other datasets prefer large $n_p$. This is because different graphs can exhibit diverse features, including node attribute features and graph topology features, which affect the sampling of positive tokens. And a large $n_p$ could introduce irrelevant nodes into positive token sequences when the features of graphs are too complex to sample relevant nodes, further hurting the performance of GCFormer.

**Analysis of $\alpha$.** To explore the influence of $\alpha$ on model performance, we vary $\alpha$ in $\{0, 0.1, \ldots, 1\}$ and observe the changes of model performance. $\alpha = 0$ or $\alpha = 1$ mean that we abandon the information from attribute-aware token sequences or topology-aware token sequences when generating the final node representations. Results across all datasets are shown in Figure 3. We can find that the optimal $\alpha$ falls in $(0, 1)$ for all datasets. This observation indicates that comprehensively considering the features of attribute and topology information is essential to learn distinguishable node representations. Another observation is that the model performances on graphs extracted from the same domain exhibit similar changing trends. For instance, GCFormer achieves the best performance when $\alpha = 0.5$ on BlogCatalog and Flickr, which are extracted from the social platforms. This may be because graphs extracted from the same domains exhibit similar graph topology features and node attribute features.

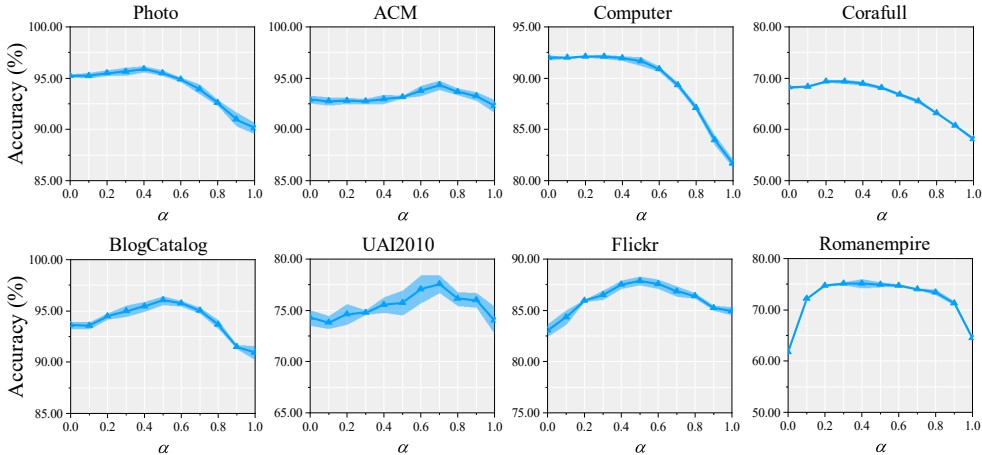

Figure 3: Performance of GCFormer with different $\alpha$ on all datasets.

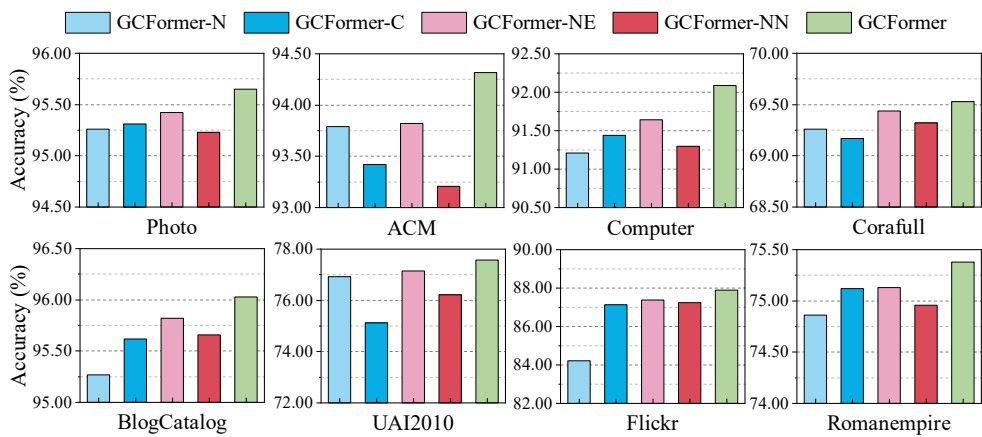

Figure 4: Performances of GCFormer and its variants.

## 5.4 Ablation Study

Generating negative token sequences and integrating contrastive learning loss are two key designs of GCFormer. To comprehensively validate the effectiveness of these designs, we propose four variants of GCFormer termed GCFormer-N, GCFormer-C, GCFormer-NE and GCFormer-NN. GCFormer-N removes the negative token sequences and the contrastive learning loss. GCFormer-C only removes the contrastive learning loss. GCFormer-NE retains the use of Transformer layers for learning negative token representations but only employs these representations in the contrastive learning loss (ignoring them in Equation 11). GCFormer-NN, conversely, directly uses the representations of negative tokens for contrastive learning without passing them through Transformer layers. We then run four variants on all datasets and the results are shown in Figure 4. We can observe that GCFormer beats four variants on all datasets, indicating the effectiveness of our key designs in enhancing the model performance. In addition, we can also find that GCFormer-C beats GCFormer-N on over half datasets. This phenomenon demonstrates that introducing negative token sequences can effectively improve the model performance. Nevertheless, the performances of GCFormer-C behind GCFormer-N on three citation networks. This situation reveals that different types of graphs can affect the gains of introducing negative tokens. In addition, The results demonstrate that GCFormer-NE outperforms GCFormer-NN on all datasets, indicating that leveraging the Transformer to learn representations of negative tokens can effectively enhance the benefits of introducing contrastive learning. Furthermore, GCFormer surpasses GCFormer-NE, suggesting that comprehensively utilizing the representations

of negative tokens through the signed aggregation operation and contrastive learning can further augment the model's ability to learn more discriminative node representations.

## 6 Conclusion

In this paper, we propose GCFormer, a novel graph Transformer for node classification. GCFormer establishes a new framework of tokenized graph Transformers to effectively learn node representations. Specifically, GCFormer develops a new hybrid token generator that generates both positive and negative token sequences. Compared to previous methods that only sample nodes with high similarity as tokens, GCFormer considers diverse nodes with high and low similarity. This merit enables GCFormer to preserve both commonality and disparity between node representations. By formulating a Transformer-based backbone and integrating contrastive learning, GCFormer can comprehensively learn distinguishable node representations from different types of token sequences. Extensive experimental results on diverse graphs extracted from different domains showcase the superiority of GCFormer in node classification compared to representative GNNs and graph Transformers.

The main limitation of GCFormer is the unified sampling strategy for different types of graphs. Experimental results show that the performance of GCFormer is sensitive to the sampling size on different graphs. The phenomenon implies that an adaptive sampling strategy is required to improve the performance and stability of GCFormer on diverse graphs.

## Acknowledgments

This work is supported by National Natural Science Foundation (62076105,U22B2017).

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

# A Detailed Experimental Settings

Here, we introduce the detailed information about experimental settings.

## A.1 Datasets

In this paper, we adopt eight datasets from diverse domains, including homophily and heterophily graphs. The statistics of datasets are summarized in Table 2.

- **Citation networks.** ACM, Corafull and UAI2010 are constructed from citation networks where nodes represent research papers and edges represent the relations between papers (*e.g.*, having common authors or citation relation).

- **Co-purchase networks.** Photo and Computer are extracted from the Amazon purchase network where nodes represent goods and edges represent that two goods appear in a same shopping list.

- **Social networks.** BlogCatalog and Flickr are generated from social platforms BlogCatalog and Flickr, respectively. Nodes represent users and edges represent social relationships between users.

- **Wikipedia.** Romanempire is extracted from English Wikipedia where nodes represent words in the text and edges represent that two words connected in the dependency tree of the sentence.

ACM, UAI2010, BlogCatalog and Flickr can be downloaded from [1]. Corafull, Photo and Computer can be downloaded from [2]. Romanempire can be downloaded from [3]. In practice, we first apply the principal components analysis (PCA) to reduce the raw features into 256-dimension vectors on Corafull, BlogCatalog, UAI2010 and Flickr since the raw features of these datasets are too sparse which waste computing resources.

Table 2: Statistics on datasets, ranked by the homophily level from high to low.

| Dataset | # nodes | # edges | # features | # labels | $H \downarrow$ |
|---|---|---|---|---|---|
| Photo | 7,650 | 238,163 | 745 | 8 | 0.83 |
| ACM | 3,025 | 1,3128 | 1,870 | 3 | 0.82 |
| Computer | 13,752 | 491,722 | 767 | 10 | 0.78 |
| Corafull | 19,793 | 126,842 | 8,710 | 70 | 0.57 |
| BlogCatalog | 5,196 | 171,743 | 8,189 | 6 | 0.40 |
| UAI2010 | 3,067 | 28,311 | 4,973 | 19 | 0.36 |
| Flickr | 7,575 | 239,738 | 12,047 | 9 | 0.24 |
| Romanempire | 22,662 | 32,927 | 300 | 18 | 0.05 |

## A.2 Parameter Configuration

Referring to the official implementations, we perform hyper-parameter tuning of baselines on each dataset. We adopt the grid search strategy to determine the optimal parameters. Specifically, We try learning rate in $\{0.001, 0.005, 0.01\}$, dropout in $\{0.3, 0.5, 0.7\}$, dimension of hidden representations in $\{128, 512\}$. For GCFormer, we try $p_k$ and $n_k$ in $\{3, 5, 7\}$, $\alpha$ in $\{0.1, \ldots, 0.9\}$, $\beta$ in $\{0.05, 0.1, 0.5, 1\}$. We implement all codes based on Python 3.8, Pytorch 1.11, and CUDA 11.0. All experiments are conducted on a Linux server with one Intel Xeon(R) Sliver 4210, 256G RAM and one RTX TITAN.

---

[1]https://github.com/zhumeiqiBUPT/AM-GCN
[2]https://github.com/JHL-HUST/NAGphormer
[3]https: //github.com/yandex-research/heterophilous-graphs

Table 3: Comparison of all models in terms of mean accuracy $\pm$ stdev (%).

| Dataset | Photo | ACM | Comuter | Corafull | BlogCatalog | UAI2010 | Flickr | Romanempire |
|---|---|---|---|---|---|---|---|---|
| $H(\mathcal{G})$ | 0.83 | 0.82 | 0.78 | 0.57 | 0.40 | 0.36 | 0.24 | 0.05 |
| ClusterSCL | $93.98_{\pm 0.43}$ | $93.27_{\pm 0.29}$ | $88.74_{\pm 0.64}$ | $62.32_{\pm 0.29}$ | $84.62_{\pm 0.1.24}$ | $74.37_{\pm 0.58}$ | $83.84_{\pm 0.42}$ | $67.37_{\pm 0.81}$ |
| CoCoS | $93.73_{\pm 0.12}$ | $93.24_{\pm 0.66}$ | $89.66_{\pm 0.48}$ | $64.25_{\pm 0.38}$ | $87.56_{\pm 0.26}$ | $75.89_{\pm 0.33}$ | $83.43_{\pm 0.59}$ | $66.28_{\pm 0.47}$ |
| NCLA | $94.21_{\pm 0.36}$ | $93.46_{\pm 0.39}$ | $89.52_{\pm 0.45}$ | $62.79_{\pm 0.34}$ | $86.69_{\pm 0.68}$ | $76.28_{\pm 0.82}$ | $84.06_{\pm 0.54}$ | $71.89_{\pm 0.49}$ |
| GCFormer | $95.65_{\pm 0.41}$ | $94.32_{\pm 0.47}$ | $92.09_{\pm 0.21}$ | $69.53_{\pm 0.35}$ | $96.03_{\pm 0.44}$ | $77.57_{\pm 0.86}$ | $87.90_{\pm 0.45}$ | $75.38_{\pm 0.68}$ |

Table 4: Performance of different GraphGPS's implementations."T" and "P" indicate the original Transformer and Performer. "L", "R" and "D" indicate the Laplacian positional encoding, RWSE structural encoding and degree-based encoding. "OOM" indicates the out-of-memory issue.

| Dataset | Photo | ACM | Comuter | Corafull | BlogCatalog | UAI2010 | Flickr | Romanempire |
|---|---|---|---|---|---|---|---|---|
| $H(\mathcal{G})$ | 0.83 | 0.82 | 0.78 | 0.57 | 0.40 | 0.36 | 0.24 | 0.05 |
| GCN+T+L | 93.79 | 93.12 | OOM | OOM | 84.62 | 74.37 | 83.84 | OOM |
| GCN+T+R | 93.81 | 93.26 | OOM | OOM | 84.62 | 74.37 | 83.84 | OOM |
| GCN+T+D | 92.95 | 92.84 | OOM | OOM | 84.62 | 74.37 | 83.84 | OOM |
| GCN+P+L | 93.74 | 93.23 | 89.21 | 61.27 | 94.21 | 75.44 | 83.54 | 68.29 |
| GCN+P+R | 93.62 | 93.31 | 89.18 | 62.08 | 94.35 | 75.14 | 82.72 | 67.52 |
| GCN+P+D | 92.38 | 92.43 | 88.06 | 59.86 | 92.75 | 70.16 | 80.88 | 64.56 |
| GCFormer | 95.65 | 94.32 | 92.09 | 69.53 | 96.03 | 77.57 | 87.90 | 75.38 |

## B    Performance Comparison with GSL-based Approaches

Here, we conduct additional experiments to validate the effectiveness of GCFormer on node classification, compared with representative graph contrastive learning-based methods. Specifically, we select three approaches, CluterSCL [39], CoCoS [43] and NCLA [31] for performance comparison. We adopt their official implementations and turn hyper-parameters accordingly on each dataset. The results are shown in Table 3. We can observe that GCFormer outperforms representative GCL-based approaches on all datasets, demonstrating its superiority in node classification.

## C    Detailed results of GraphGPS

Here, we provide the detailed results of different implementations of GraphGPS. We adopt this resource code[1] for experiments. The results are shown in Table 4. The results demonstrate that GCFormer outperforms GraphGPS on all datasets, highlighting the effectiveness of GCFormer in comparison to representative graph Transformers in the task of node classification.

---

[1]https://github.com/luis-mueller/probing-graph-transformers

