# OpenReview forum: "Leveraging Contrastive Learning for Enhanced Node Representations in Tokenized Graph Transformers"
_NeurIPS.cc/2024/Conference — NeurIPS 2024 poster_

### Official Review · Reviewer_pga6 · 2024-07-05

**Soundness:** 2
**Presentation:** 2
**Contribution:** 3
**Rating:** 6
**Confidence:** 4

**Summary:**

In this work, the authors present GCFormer, a new graph transformer (GT) architecture for improved node classification on homophilic and heterophilic graphs. To this end, the authors propose to sample a fixed number of both positive and negative tokens for each node $v$ in the graph and then restrict the attention computation for $v$ to only the positive and negative tokens. The sampling procedure is based on similarity scores that are computed both based on node attributes as well as topological information based on powers of the normalized adjacency matrix. The authors further introduce a contrastive loss, added to the supervised cross-entropy loss, with the goal of promoting  the representation of a given node to be close to the representation of the positive samples and to be far from the representation of the negative samples. GCFormer is evaluated on a range of node classification tasks, with varying levels of homophily.

**Strengths:**

The strengths of this work lie in the depth of the experimental study:

- The authors select eight datasets with a wide range of homophily levels.
- The authors tune all baselines with the same hyper-parameter ranges, enabling a fair comparison to prior works.
- The authors present an ablation study, indicating the effectiveness of their proposed approach. Specifically, it becomes apparent that the idea of learning representations close to positive tokens and far away from negative tokens is effective (which, to me, is the central innovation in this work).

**Weaknesses:**

The main weakness of this paper is its poor disambiguation from prior work and the resulting weakened motivation of the proposed model. Concretely, it becomes evident in L51-57, as well as the related work, that the authors base their work mainly on GTs that, for each node $v$, perform attention over a reduced set of nodes similar to $v$ (based on node attributes and topological information) — I will refer to this class of models as *positive-sampling GTs*. The authors cite the following works:

- ANS-GT (https://arxiv.org/abs/2210.03930)
- Gophormer (https://arxiv.org/abs/2110.13094)
- VCR-Graphormer (https://arxiv.org/abs/2403.16030)
- NAGphormer (https://arxiv.org/abs/2305.12677)

First I want to note that out of these, the authors *only compare to ANS-GT and NAGphormer*. The other transformer baselines in the experimental section are

- SGFormer (https://arxiv.org/abs/2306.10759)
- Specformer (https://arxiv.org/abs/2303.01028)

none of which fall into the category of positive-sampling GTs and instead perform attention between all node pairs. In addition, other GTs that perform attention between all node pairs are missing in the experimental study, most notably GraphGPS with linear attention (https://arxiv.org/abs/2205.12454). Specifically, the authors claim that GraphGPS has limited modeling capacity, yet the authors do not compare their approach to GraphGPS, neither do they present any other evidence supporting this claim. Instead, the authors argue that the design of GraphGPS "could restrict the modeling capacity of Transformer and further limit the model performance” (L93-94). Indeed, GraphGPS with Performer attention appears to show much better performance on Roman-Empire than GCFormer and any of the baselines in this work; see Müller et al. (https://arxiv.org/abs/2302.04181) for GraphGPS+Performer on Roman-Empire, and see my question below on whether these results are directly comparable.

*This is what I mean with poor disambiguation from prior work*. It is evident that the authors want to improve the state-of-the-art of positive-sampling GTs but the authors make claims throughout the paper that are referring to graph transformers in general. For example, in L61-63: “[…] existing graph Transformers can not comprehensively utilize both similar and dissimilar nodes to learn the representation of the target node, inevitably limiting the model performance for node classification”. Without additional evidence, this statement only holds for positive-sampling GTs. It is not obvious that the same holds for GTs performing attention between all node pairs, such as SGFormer, Specformer or GraphGPS with linear attention.

As a result, the work is not well motivated. While the motivation of GCFormer with respect to positive-sampling GTs is given, it is not demonstrated or argued sufficiently what the short-comings of models such as SGFormer, Specformer or GraphGPS with linear attention are in the first place and why GCFormer (or any of the positive-sampling GTs that GCFormer improves over) offers a conceptual improvement.

The only remedy to this are the empirical results, where GCFormer comes out on top compared to the positive-sampling GTs, SGFormer and Specformer. However, on five out of eight datasets, the second best performance is *within standard deviation* of GCFormer, indicating small but not convincing improvements over existing methods. Most notably, at the cost of less generality, since GCFormer does not compute attention between all node pairs.

I expect the authors to clearly motivate their approach and demonstrate the benefits GCFormer offers. In addition, I hope that the authors can provide supporting evidence to the claims made about GraphGPS, for example by including GraphGPS with linear attention as a baseline in the experimental section or by arguing more specifically about its shortcomings in the context of node classification.

**Questions:**

- Is the empirical setting on Roman-Empire derived from Platanov et al. (https://arxiv.org/abs/2302.11640) and if not, what changes to the empirical setting did the authors make? Are the results of GCFormer comparable to the results in Müller et al. (https://arxiv.org/abs/2302.04181)?
- Are the $V^{a,p}_i$ in L180 and $N^{a,p}_i$ in L188 the same? I suspect this to be a typo but I wanted to make sure I did not misunderstand how the authors construct the positive and negative token sequences from the sampled positive and negative similarity scores.

**Limitations:**

The limitations of the work were adequately addressed.

---

> ### Author Rebuttal · Authors · 2024-08-07
>
> Thank you for the detailed comments and valuable questions. We provide details to clarify your major concerns.
>
> >**Q1.** I expect the authors to clearly motivate their approach and demonstrate the benefits GCFormer offers. In addition, I hope that the authors can provide supporting evidence to the claims made about GraphGPS, for example by including GraphGPS with linear attention as a baseline in the experimental section or by arguing more specifically about its shortcomings in the context of node classification.
>
> **A1.** Thank you for your detailed and insightful comments. Firstly, we discuss the potential limitations of linear attention-based graph Transformers in the context of node classification tasks. These approaches require performing the attention calculation on all node pairs, which can lead to what is known as the "over-globalizing" problem. A recent study [1] (not included in the original version as it was released in May) provides both empirical evidence and theoretical analysis to show that calculating attention scores for all nodes can cause the model to overly rely on global information, which can negatively affect the model's performance in node classification tasks. In contrast, tokenized graph Transformers, such as GCFormer, only calculate attention scores between tokens within the token sequence, naturally avoiding the over-globalizing issue.
>
> Secondly, the claim that "existing graph Transformers cannot comprehensively utilize both similar and dissimilar nodes to learn the representation of the target node" is supported by the following observations. Existing tokenized graph Transformers typically ignore dissimilar nodes when learning node representations. For linear attention-based approaches, such as GraphGPS and SGFormer, the calculation of the self-attention mechanism can be seen as conducting a single aggregation operation on a fully connected graph. Each node receives information from all other nodes, including both similar and dissimilar ones. However, the aggregation weights produced by the self-attention mechanism are always positive. Utilizing positive weights to aggregate information from dissimilar nodes can be viewed as a smoothing operation [2]. This situation suggests that linear attention-based approaches are inefficient in utilizing both similar and dissimilar nodes to learn the representation of the target node, potentially leading to suboptimal performance.
>
> Thirdly, to further enhance the performance of graph Transformers for node classification, we propose GCFormer, which is built upon the tokenized graph Transformer architecture. The main contribution of GCFormer lies in developing a new architecture for graph Transformers that enables the model to comprehensively leverage various tokens containing both similar and dissimilar nodes to learn node representations. By introducing negative tokens, GCFormer is the first attempt to explicitly incorporate dissimilar nodes into the learning process, thereby enriching the representations learned by graph Transformers. This innovation allows GCFormer to effectively distinguish between similar and dissimilar nodes, leading to improved performance in node classification tasks.
>
> Finally, we evaluate the performance of GraphGPS on our datasets. We utilize the implementation of GraphGPS provided by [3]. Note that the authors do not provide a ready-to-use script to reproduce the results on five new datasets, including Roman-empire. Therefore, we use the scripts for node classification on small graphs provided by [3] to search for the optimal combination of parameters. The results are summarized in the **common response** due to the space limitation.
> Based on the existing results, we can observe that GCFormer outperforms GraphGPS in node classification tasks.
> We will add the above discussions and results to the revised version. Thank you.
>
> >**Q2.** Is the empirical setting on Roman-Empire derived from Platanov et al. and if not, what changes to the empirical setting did the authors make? Are the results of GCFormer comparable to the results in Müller et al.?
>
> **A2.** No, we resplit all datasets via the same random seed in our experimental environment to enable a fair comparison. And we cannot reproduce the results of GraphGPS reported in Table 4 [3] since the authors do not make the code of this part available in their official repertory. To address your concerns, we evaluate the performance of GCFormer on Question, which is the largest one of five new datasets. The results are reported in the following table:
> |                          | Questions |
> |--------------------------|-----------|
> | GCN+T+LPE | 77.85     |
> | GCN+T+RWSE  | 76.45     |
> | GCN+T+DEG  | 74.24     |
> | GCN+P+LPE | 76.71     |
> | GCN+P+RWSE  | 77.14     |
> | GCN+P+DEG   | 76.51     |
> | GCFormer                 | 80.87     |
>
> We can observe that GCFormer outperforms all implementations of GPS on the largest dataset, demonstrating the effectiveness of GCFormer in node classification.
> We will add the above results and discussions to the revised version.
>
> >**Q3.** Questions about $V_{i}^{a,p}$ and $N^{a,p}_{i}$.
>
> **A3.** Thank you for pointing out this. They are the same. We will fix these typos in the revised version.
>
> [1] Yujie Xing, et al. Less is More: on the Over-Globalizing Problem in Graph Transformers. ICML 2024.
>
> [2] Deyu Bo, et al. Beyond Low-Frequency Information in Graph Convolutional Networks. AAAI 2021.
>
> [3] Luis Müller, et al. Attending to Graph Transformers. TMLR, 2024.

---

> > ### Comment · Reviewer_pga6 · 2024-08-07
> >
> > I thank the authors for their rebuttal.
> >
> > ### Questions
> > First, the authors say that
> >
> > > Existing tokenized graph Transformers typically ignore dissimilar nodes when learning node representations
> > >
> >
> > Can the authors clearly define which class of architectures the authors are referring to with "tokenized graph Transformers"? Are those the class of architectures that I referred to as "positive-sampling GTs"? If so, I find the authors' terminology misleading, and could potentially be confused with e.g., TokenGT (https://arxiv.org/abs/2207.02505).
> >
> > Next, the authors state
> >
> > > Utilizing positive weights to aggregate information from dissimilar nodes can be viewed as a smoothing operation
> > >
> >
> > I would imagine that a multi-head attention mechanism could independently learn to pay attention to both similar and dissimilar nodes via two separate attention heads, respectively. Whether or not existing graph transformers do so effectively is up for debate. However, I still do not see how the authors are providing convincing argumentation for this claim.
> >
> > ### New empirical results
> > I appreciate the authors time and effort in providing additional empirical results, comparing to GraphGPS (with full- and Performer-attention) on a variety of datasets. This definitely strengthens the claims made in the paper.
> >
> > ### Summary
> > I will raise my score due to the additional empirical evidence that GCFormer can outperform standard graph transformers such as GraphGPS on a variety of datasets. I am still, however, concerned about the way the authors frame their contribution. In my view, the authors improve upon a particular class of architectures (graph transformers that have improved scalability by computing attention, for each query nodes only to a subset of key nodes) but make claims about graph transformers in general that are not supported by theoretical or empirical evidence. For example, it is not clear that the reason why SGFormer or GraphGPS perform worse on the presented datasets is due to the explanation given by the authors in the rebuttal:
> >
> > >  For linear attention-based approaches, such as GraphGPS and SGFormer, the calculation of the self-attention mechanism can be seen as conducting a single aggregation operation on a fully connected graph. Each node receives information from all other nodes, including both similar and dissimilar ones. However, the aggregation weights produced by the self-attention mechanism are always positive. Utilizing positive weights to aggregate information from dissimilar nodes can be viewed as a smoothing operation [2]. This situation suggests that linear attention-based approaches are inefficient in utilizing both similar and dissimilar nodes to learn the representation of the target node, potentially leading to suboptimal performance.
> > >

---

> > > ### Author Response · Authors · 2024-08-08
> > > **Responses for Reviewer pga6**
> > >
> > > We are very grateful for the reviewer's prompt and positive response. We answer your questions as follows.
> > >
> > > >**Q1.** Can the authors clearly define which class of architectures the authors are referring to with "tokenized graph Transformers"? Are those the class of architectures that I referred to as "positive-sampling GTs"? If so, I find the authors' terminology misleading, and could potentially be confused with e.g., TokenGT (https://arxiv.org/abs/2207.02505).
> > >
> > > **A1.**  In our paper, "tokenized graph Transformers" refer to approaches that initially assign each node with token sequences as the model input and then utilize the Transformer architecture to learn node representations for node classification tasks. Following your suggestion, we will clarify the definition of "tokenized graph Transformer" and emphasize the distinctions from existing methods, such as GraphGPS and TokenGT, to avoid any potential misinterpretation in the revised version. We appreciate your feedback.
> > >
> > > >**Q2.** Utilizing positive weights to aggregate information from dissimilar nodes can be viewed as a smoothing operation.
> > >
> > > **A2.** The self-attention mechanism can be viewed as aggregating information from all node pairs. While a multi-head attention mechanism can independently learn to focus on both similar and dissimilar nodes via separate attention heads, it is important to note that the self-attention mechanism inherently generates only positive attention weights. This means that similar nodes and dissimilar nodes are typically assigned higher and lower attention values, respectively. However, recent studies [1] have shown that leveraging only positive aggregation weights can limit the model to preserving primarily low-frequency information. Furthermore, another study [2] suggests that the Transformer architecture can be seen as a low-pass filter, supporting this perspective.
> > > Therefore, relying solely on positive aggregation weights for aggregation may not be sufficient. An ideal approach would involve using both positive and negative aggregation weights to aggregate the information from similar and dissimilar nodes, respectively. By doing so, both low-frequency and high-frequency information on the graph can be effectively preserved for learning node representations.
> > >
> > > We will add the above discussion to the revised version. Thank you.
> > >
> > > >**Q3.** About the contributions of GCFormer.
> > >
> > > **A3.** Following your suggestion and as discussed in **A1**, we will clarify in the revised version that GCFormer specifically enhances the performance of tokenized graph Transformers, rather than graph Transformers in general, to ensure a clear and accurate description.
> > >
> > > [1] Deyu Bo, et al. Beyond Low-Frequency Information in Graph Convolutional Networks. AAAI 2021.
> > >
> > > [2] Peihao Wang, et al. Anti-Oversmoothing in Deep Vision Transformers via the Fourier Domain Analysis: From Theory to Practice. ICLR. 2021.

---

> > > > ### Comment · Reviewer_pga6 · 2024-08-10
> > > >
> > > > I thank the authors for their response.
> > > >
> > > > I have decided to raise my score, in light of the convincing empirical results and with the hope that the authors improve their discussion of related work, as suggested above.

---

> > > > > ### Author Response · Authors · 2024-08-11
> > > > >
> > > > > We are very grateful for your patient discussion and insightful comments, which are very helpful in improving our paper. We will carefully prepare the final version of the paper based on the content discussed with you.

---

### Official Review · Reviewer_9KjY · 2024-07-08

**Soundness:** 3
**Presentation:** 3
**Contribution:** 3
**Rating:** 8
**Confidence:** 4

**Summary:**

This paper proposes a new graph Transformers for node classification. Different from previous graph Transformers that only select nodes with high-similarity to construct the token sequences, this paper proposes GCFormer, that considers both high-similairity and low-similarity nodes as tokens. Moreover, GCFormer introduces the contrastive learning to further enhance the quality of learned node representations.
Overall, the idea that introduces negative token sequences and leverages contrastive learning to enhance the performance of tokenized graph Transformers is interesting.

**Strengths:**

1. This paper is written and easy to follow.
2. This paper proposes a new architecture of graph Transformer.
3. The empirical performance of GCFormer is outstanding.

**Weaknesses:**

1. Some representative baselines are missing.
2. More details of GCFormer are required.

**Questions:**

1. I suggest authors add recent representative graph Transformers, such as VCR-Graphormer [1], as baselines to make the experimental results more convicing.
2. How do you utilize the virtual nodes to learn representations of negative token sequences? Do you assign learnable features to each virtual nodes, (like CLS tokens)?
3. Moreover, how do you construct the token sequences (both positive and negative)? It seems that you set the target node and the virtual node as the first items of the positive token sequences and negative token sequences, respectively.   I suggest the authors provide more detailed information about the construction of token sequences.

[1] Fu D, Hua Z, Xie Y, et al. VCR-graphormer: A mini-batch graph transformer via virtual connections. ICLR 2024.

**Limitations:**

The authors have discussed the limitations of the proposed method.

---

> ### Author Rebuttal · Authors · 2024-08-07
>
> Many thanks for the positive evaluation and insightful questions that help us improve this work. We provide the following detailed responses to your questions.
>
> > **Q1.** I suggest authors add recent representative graph Transformers, such as VCR-Graphormer [1], as baselines to make the experimental results more convicing.
>
> **A1.** Per your suggestion, we conduct additional experiments to evaluate the performance of VCR-Graphormer [1] via the official implementation on GitHub. The results are as follows:
> |                | Pho.  | ACM   | Com.  | Cor.  | Blo.  | UAI.  | Fli.  | Rom.  |
> |----------------|-------|-------|-------|-------|-------|-------|-------|-------|
> | VCR-Graphormer | 95.13 | 93.24 | 90.14 | 68.96 | 93.92 | 75.78 | 86.23 | 74.76 |
> | GCFormer       | 95.65 | 94.32 | 92.09 | 69.53 | 96.03 | 77.57 | 87.90 | 75.38 |
>
> We can observe that GCFormer consistently surpasses VCR-Graphormer on all datasets, which demonstrates the superiority of GCFormer compared to recent tokenized graph Transformers.
> We will add the above results and discussions to the revised version.
>
> > **Q2.** How do you utilize the virtual nodes to learn representations of negative token sequences? Do you assign learnable features to each virtual nodes, (like CLS tokens)?
>
> **A2.** Thank you for your attention on this point. Each virtual node is associated with a learnable feature vector. Through the Transformer layer, these virtual nodes can preserve the information of tokens in negative token sequences via the self-attention mechanism. This usage of virtual nodes is analogous to the use of CLS tokens in natural language processing, where the CLS token aggregates information from the entire sequence to perform tasks such as text classification. In the context of GCFormer, the virtual nodes serve a similar purpose, aggregating and preserving the information from the negative tokens, which is then used to enhance the contrastive learning process.
> We will reorganize the related sentences for a clear description in the revised version.
>
> > **Q3.** Moreover, how do you construct the token sequences (both positive and negative)? It seems that you set the target node and the virtual node as the first items of the positive token sequences and negative token sequences, respectively. I suggest the authors provide more detailed information about the construction of token sequences.
>
> **A3.** Thank you for your helpful suggestion. In practice, we utilize the target node and it's positive sampling nodes to construct the positive token sequence where the target node is the first item in the sequence. Similarly, we use the virtual node and the target node's negative sampling nodes to construct the negative token sequence where the virtual node is the first item. By placing the target node at the beginning of the positive sequence and the virtual node at the beginning of the negative sequence, we ensure that the first item in each sequence captures the learned information from the corresponding input sequence. This allows us to leverage the first item of each sequence effectively for downstream tasks.
> We will update the description of the construction of positive and negative token sequences in the revised version.
>
> [1] Dongqi Fu, et al. VCR-Graphormer: A Mini-batch Graph Transformer via Virtual Connections. ICLR 2024.

---

> > ### Comment · Reviewer_9KjY · 2024-08-08
> >
> > Thank you for addressing my concerns. After carefully reviewing your responses to my comments and those of other reviewers, I have increased my score. In my opinion, GCFormer is a simple, intuitive, and effective method that enhances the performance of token sequence-based GTs. I hope that you can incorporate the discussions from your rebuttal, particularly the experimental results, into the final version.

---

> > > ### Author Response · Authors · 2024-08-08
> > >
> > > Many thanks for your positive feedback that greatly encourages us. We will carefully revise the paper according to the reviewers' suggestions and additional experimental results.

---

### Official Review · Reviewer_rUu7 · 2024-07-10

**Soundness:** 2
**Presentation:** 3
**Contribution:** 2
**Rating:** 5
**Confidence:** 4

**Summary:**

This paper proposes a token sequence-based graph transformer method named Graph Contrastive Transformer (GCFormer). They first sample top-$k$ similarity nodes as positive token sequence and regard the rest as negative token sequence. Then, they use transformer to obtain the positive and negative representations and subtract negative representation from positive representation to get the node representation. Finally, the combination of contrastive loss and cross entropy loss is taken to optimize the model.

**Strengths:**

- The overall paper is clear and easy to understand. The introduction of proposed method and discussion on experiment is detailed.
- The idea of introducing negative samples to graph transformer seems promising.

**Weaknesses:**

- The method is not new to the community and the intuition behind getting node representation by subtracting negative embedding from positive embedding is not clear.
- Experimental result is not solid enough. Graph contrastive learning methods should be included as baselines. Figure 2 does not provide insightful information regarding the choose of $p_k$ and $n_k$.

**Questions:**

- In the ablation study shown in Figure 4, GCNFormer-N scenario can achieve better performance than GCNFormer-C in some cases, which shows the proposed negative token sequence might harm the model performance. But introducing contrastive loss can remedy this problem. Could the authors provide some explanation? Have the authors tried using CE loss without negative samples version + CL loss?

---

> ### Author Rebuttal · Authors · 2024-08-07
>
> We appreciate the reviewer for providing valuable feedback and comments on our paper. Following are our detailed responses to your questions.
> >**Q1.** The method is not new to the community and the intuition behind getting node representation by subtracting negative embedding from positive embedding is not clear.
>
> **A1.** Previous research has focused on integrating contrastive learning into GNNs to improve performance in supervised node classification tasks. However, GNNs and graph Transformers represent distinct methodologies. As discussed in Section 2.2, strategies effective in GNNs may not directly apply to graph Transformers. This paper addresses the gap in leveraging contrastive learning for graph Transformers by proposing GCFormer, which introduces a novel hybrid token generator to produce both positive and negative tokens for each target node. Building on this capability, we formulate a contrastive learning-based loss function that fully exploits the information learned from both positive and negative tokens, thereby enhancing the overall model performance.
>
> Inspired by the recent study FAGCN [1], which leverages positive and negative weights to aggregate low-frequency and high-frequency information, respectively, GCFormer employs a subtraction operation to mimic the signed aggregation operation. This enables the comprehensive learning of node representations from different types of tokens as discussed in Line 208-213.
> We will add the above discussions to the revised version to highlight the novelty and motivation of the proposed GCFormer.
>
> >**Q2.** Experimental result is not solid enough. Graph contrastive learning methods should be included as baselines. Figure 2 does not provide insightful information regarding the choice of $p_k$ and $n_k$.
>
> **A2.** Thank you for your helpful suggestions. To address your concerns, we select three recent graph contrastive learning-based approaches for supervised node classification as baselines, CluterSCL[2], CoCoS[3], and NCLA[4]. The results are reported in the **common response** due to the space limitation.
> The experimental results indicate the superiority of GCFormer for node classification, compared to representative graph contrastive learning-based methods.
>
> Figure 2 illustrates the impact of the sampling sizes $n_k$​ and $p_k$​ on the model's accuracy across various datasets. High accuracy is denoted by blue regions, while low accuracy is indicated by red regions. For $n_k$​, we observe that for most datasets (except ACM), the blue regions typically appear in the upper half of the grid along the y-axis, suggesting that a larger value of $n_k$ tends to lead to higher accuracy. For $p_k$​, though the distribution of the blue areas is less consistent, over half of datasets require a small value of $p_k$ to achieve the satisfied performance. These observations suggest that the combination of a large value of $n_k$​ and a small value of $p_k$ could be effective for achieving competitive performance.
>
> We will add the above results and discussions to the revised version to strengthen the experiment section.
>
>
> >**Q3.** In the ablation study shown in Figure 4, GCNFormer-N scenario can achieve better performance than GCNFormer-C in some cases, which shows the proposed negative token sequence might harm the model performance. But introducing contrastive loss can remedy this problem. Could the authors provide some explanation? Have the authors tried using CE loss without negative samples version + CL loss?
>
> **A3.** Thank you for your insightful comments. GCFormer-C only encourages the target node's representation to be distinct from the virtual node's representation extracted from negative tokens, with no strong constraints imposed on the relationships among the representations of the target node, positive tokens, and negative tokens. In contrast, the contrastive loss function, as described in Equation (13), explicitly narrows the distance between the target node's representation and the central representation of positive tokens while simultaneously enlarging the distance between the target node's representation and the representations of negative tokens. This loss function enhances the model's ability to learn more distinguished representations of the target node, positive tokens, and negative tokens, which in turn further improves the overall model performance.
>
> To address the concerns raised and provide a more comprehensive evaluation, we introduce two additional variants of GCFormer: GCFormer-NN and GCFormer-NL. GCFormer-NN retains the use of Transformer layers for learning negative token representations but only employs these representations in the contrastive learning loss (ignoring them in Equation 11). GCFormer-NL, conversely, directly uses the representations of negative tokens for contrastive learning without passing them through Transformer layers. The results of these variants are also summarized in the common response.
>
> We can observe that utilizing the representations of negative tokens learned by Tranformer for contrastive learning can consistently improve the model performance. Moreover, GCFormer beats GCFormer-NE, indicating that combining Eq. (11) with the contrastive learning loss can fully extract the information of negative tokens to learn distinguished node representations.
> We will add the above results and discussions to the revised version. Thank you.
>
>
> [1] Deyu Bo, et al. Beyond Low-Frequency Information in Graph Convolutional Networks. AAAI 2021.
>
> [2] Yanling Wang, et al. ClusterSCL: Cluster-aware Supervised Contrastive Learning on Graphs. The Web Conference 2022.
>
> [3] Siyue Xie, et al. CoCoS: Enhancing Semi-Supervised Learning on Graphs with Unlabeled Data via Contrastive Context Sharing. AAAI 2022.
>
> [4] Xiao Shen, et al. Neighbor Contrastive Learning on Learnable Graph Augmentation. AAAI 2023.

---

> > ### Comment · Reviewer_rUu7 · 2024-08-10
> >
> > Most of my concerns have been addressed by the additional experimental results. I will raise my score, and hope the author can include these comparisons and discussions in your revised paper.

---

> > > ### Author Response · Authors · 2024-08-11
> > >
> > > We are very grateful for your positive feedback and will carefully prepare the final version of our paper based on the experimental results and discussions provided in the rebuttal.

---

### Author Rebuttal · Authors · 2024-08-07

Here, we present the experimental results to address the concerns of **Reviewer rUu7** and **Reviewer pga6**.

### **Reviewer rUu7**
we first report the performance of GCFormer and three representative GCL-based methods for supervised node classification. The results are as follows:
|            | Pho.  | ACM   | Com.  | Cor.  | Blo.  | UAI.  | Fli.  | Rom.  |
|------------|-------|-------|-------|-------|-------|-------|-------|-------|
| ClusterSCL | 93.98 | 93.27 | 88.74 | 62.32 | 84.62 | 74.37 | 83.84 | 67.37 |
| CoCoS      | 93.73 | 93.24 | 89.66 | 64.25 | 87.56 | 75.89 | 83.43 | 66.28 |
| NCLA       | 94.21 | 93.46 | 89.52 | 62.79 | 86.69 | 76.28 | 84.06 | 71.89 |
| GCFormer   | 95.65 | 94.32 | 92.09 | 69.53 | 96.03 | 77.57 | 87.90 | 75.38 |

We can observe that GCFormer outperforms representative GCL-based approaches on all datasets, demonstrating its superiority in node classification.



Then, the results of GCFormer  and its four variants are as follows:
|             | Pho.  | ACM   | Com.  | Cor.  | Blo.  | UAI.  | Fli.  | Rom.  |
|-------------|-------|-------|-------|-------|-------|-------|-------|-------|
| GCFormer-N  | 95.26 | 93.79 | 91.21 | 69.26 | 95.27 | 76.93 | 84.21 | 74.86 |
| GCFormer-C  | 95.31 | 93.42 | 91.44 | 69.17 | 95.62 | 75.13 | 87.14 | 75.12 |
| GCFormer-NE | 95.42 | 93.82 | 91.64 | 69.44 | 95.82 | 77.14 | 87.37 | 75.13 |
| GCFormer-NN | 95.23 | 93.21 | 91.30 | 69.32 | 95.66 | 76.22 | 87.23 | 74.96 |
| GCFormer    | 95.65 | 94.32 | 92.09 | 69.53 | 96.03 | 77.57 | 87.90 | 75.38 |

The results demonstrate that GCFormer-NE outperforms GCFormer-NN on all datasets, indicating that leveraging the Transformer to learn representations of negative tokens can effectively enhance the benefits of introducing contrastive learning. Furthermore, GCFormer surpasses GCFormer-NE, suggesting that comprehensively utilizing the representations of negative tokens through the signed aggregation operation and contrastive learning can further augment the model's ability to learn more discriminative node representations.



### **Reviewer pga6**
We report the performance of GraphGPS implemented by different model combinations. The results are as follows:
|            | Pho.  | ACM   | Com.  | Cor.  | Blo.  | UAI.  | Fli.  | Rom.  |
|------------|-------|-------|-------|-------|-------|-------|-------|-------|
| GCN+Transformer+LPE  | 93.79 | 93.12 | OOM   | OOM   | 94.14 | 74.06 | 83.61 | OOM   |
| GCN+Transformer+RWSE | 93.81 | 93.26 | OOM   | OOM   | 94.25 | 75.39 | 82.38 | OOM   |
| GCN+Transformer+DEG  | 92.95 | 92.84 | OOM   | OOM   | 92.51 | 69.81 | 81.54 | OOM   |
| GCN+Performer+LPE  | 93.74 | 93.23 | 89.21 | 61.27 | 94.21 | 75.44 | 83.54 | 68.29 |
| GCN+Performer+RWSE | 93.62 | 93.31 | 89.18 | 62.08 | 94.35 | 75.14 | 82.72 | 67.52 |
| GCN+Performer+DEG  | 92.38 | 92.43 | 88.06 | 59.86 | 92.75 | 70.16 | 80.88 | 64.56 |
| GCFormer   | 95.65 | 94.32 | 92.09 | 69.53 | 96.03 | 77.57 | 87.90  | 75.38 |

The results demonstrate that GCFormer outperforms GraphGPS on all datasets, highlighting the effectiveness of GCFormer in comparison to representative graph Transformers in the task of node classification.


**We will incorporate all the above results and discussions in the revised version, and we appreciate the reviewers' valuable suggestions.**

---

### Decision · Program_Chairs · 2024-09-25

**Decision:**

Accept (poster)

**Comment:**

This paper proposes a new graph Transformers for node classification, named Graph Contrastive Transformer (GCFormer). Different from previous graph Transformers that only select nodes with high-similarity to construct the token sequences, GCFormer considers both high-similairity and low-similarity nodes as tokens, and introduces contrastive learning to enhance the quality of learned node representations. The work provides deep empirical studies, showing strong results on eight datasets with a wide range of homophily levels and rich ablation studies. It’d be beneficial to include more comparison with representative baselines, such as VCR-Graphormers and other Graph contrastive learning methods.